# A Weakly Supervised and Globally Explainable Learning Framework for Brain Tumor Segmentation

## Abstract

Brain tumors are a prevalent clinical disease that causes significant suffering for patients. Machine-based segmentation of brain tumors can assist doctors in diagnosis and providing better treatment. However, the complex structure of brain tumors presents a challenge for automatic tumor detection. Deep learning techniques have shown great potential in learning feature representations, but they often require a large number of samples with pixel-level annotations for training for implementing objects segmentation. Additionally, the lack of interpretability in deep learning models hinders their application in medical scenarios. In this paper, we propose a counterfactual generation framework that not only achieves exceptional performance in brain tumor segmentation without the need for pixel-level annotations, but also provides explainability. Our framework effectively separate class-related features from class-unrelated features of the samples, and generate new samples that preserve identity features while altering class attributes by embedding different class-related features. We can accurately identify tumor regions through performing comparison between original abnormal images and generated normal samples which preserve original identity features. We employ topological data analysis for projecting extracted class-related features into a globally explainable class-related manifold. Furthermore, by actively manipulating the generation of images with different class attributes with defined paths, we can provide a more comprehensive and robust explanation of the model. We evaluate our proposed method through experiments conducted on two datasets, which demonstrates superior performance of brain segmentation.

## 1 Introduction

Brain tumors refer to an abnormal growth or mass of cells in the brain, which can be either cancerous (malignant) or non-cancerous (benign). Brain tumors can cause various neurological symptoms, such as headaches, seizures, cognitive impairments, and more. In some cases, they can lead to intracranial hemorrhage, resulting in sudden and severe symptoms like loss of consciousness, paralysis, and potentially life-threatening complications. The global prevalence of brain tumors is a significant concern, prompting researchers and medical professionals to focus on their detection and treatment.

Diagnosing brain tumors entirely by humans faces challenges due to irregular shape and size, as well as the poor contrast and blurred boundaries of tumor tissues, besides, the existence of a large number of patients puts enormous pressure on the already scarce medical resources. Therefore, accurate segmentation of brain tumors using machines can greatly assist doctors in diagnosing and providing improved treatments for patients. Numerous studies have investigated the segmentation of brain tumors using MRI images, including traditional machine learning algorithms, such as random forest Koley et al. (2016); Goetz et al. (2015); Soltaninejad et al. (2018), support vector machine Kumar et al. (2017); Kharrat et al. (2010), and deep learning algorithms like fully convolutional networks Sun et al. (2021), cascaded CNNs Wang et al. (2018; 2019); Jiang et al. (2020); Ranjbarzadeh et al. (2021), and dilated/atrous convolutions Chen et al. (2019a); Yang et al. (2020); Cahall et al. (2021). Some researchers have also proposed brain tumor segmentation algorithms based on top-down/bottom-up strategies Zhang et al. (2021); Guan et al. (2022); Rehman et al. (2021); Jiang

et al. (2022). However, all these methods require pixel-level semantic annotations, which is both expensive and challenging to implement in clinical settings, especially for complex diseases like brain tumors.

There are some brain tumor segmentation methods without the relying of pixel-level annotations, including thresholding-based methods Nyo et al. (2022); Khilkhal & Ismael (2022) and clustering-based segmentation methods Alam et al. (2019); Setyawan et al. (2018). However, these traditional approaches have limited accuracy and are prone to interference due to weak ability in complex features extraction. Deep learning has demonstrated remarkable capability in extracting and representing complex features. In recent years, some deep learning based methods that only rely on image-level class labels and do not require pixel-level annotation for training for objects localization/segmentation have been proposed, which are known as weakly supervised object localization (WSOL) Choe & Shim (2019); Xue et al. (2019); Pan et al. (2021). The most commonly used technique for WSOL is the class activation map (CAM) Zhou et al. (2016), which generates semantic-aware localization maps by utilizing the activation map from the final convolution layer to estimate object bounding boxes. However, CAM tends to underestimate object regions as it only identifies the most discriminative parts of an object, rather than the entire object area, resulting in poor localization performance. To address this issue, Gao et al. proposed the token semantic coupled attention map (TS-CAM) approach using visual transformer Dosovitskiy et al. (2020), which splits the image into multiple patch tokens and makes these tokens aware of object categories for more accurate object localization Yao et al. (2022). However, splitting images into patches may neglect the spatial coherence of objects, making it difficult to predict complete activation. To overcome this limitation, Bai et al. introduced the Spatial Calibration Module (SCM), an external module designed for Transformers, to produce activation maps with sharper boundaries Bai et al. (2022). Kim et al. improved the alignment of feature directions in the entire object region with class-specific weights to bridge the gap between classification and localization and expand the activated region to cover the entire object area Kim et al. (2022). Tao et al. obtained pseudo lesion segmentation maps based on CAM and used them to mask the original abnormal and generated healthy images. By minimizing the distances between the marked generated normal images and the marked original abnormal images, more accurate segmentation maps were guided for generating Tao et al. (2023). These methods improve the accuracy of object localization or segmentation compared to CAM. However, due to the complex interactions between features and context, local gradients in these methods are susceptible to local traps, resulting in biased Shrikumar et al. (2017), unstable Adebayo et al. (2018), inaccurate or even misleading Chen et al. (2019b); Samek et al. (2019) localization results. Additionally, persistent criticisms Ghassemi et al. (2021); Rudin (2019) that highlight feature associations provide limited semantic knowledge about the decision rules. Besides, these methods lack global explainability, making their widespread application in medical scenarios more challenging.

In this paper, we present a weakly supervised learning framework that effectively addresses the challenges mentioned above. Our framework eliminates the relying of pixel-level annotations in the segmentation task and achieves explainable learning. Specifically, we propose a class association embedding (CAE) framework, which includes a symmetrical cyclic generation adversarial network and a Paired Random Shuffle Training (PRST) scheme. This framework is designed to embed class associations and transform the class style of brain images while preserving individual identity features. The symmetrical network consists of one encoder with two sub-modules, one decoder, and one multi-class discriminator. One module in the encoder is responsible for extracting class-related features, generating class-style (CS) codes within a unified domain. The other module focuses on extracting class-unrelated features, acquiring individual-style (IS) codes. By performing topological data analysis on learned class-related features, we can thoroughly explore the global explainable rules that govern the entire dataset and the relationships between samples in one distilled unified manifold, which enables us to identify meaningful and rule-based pathways for class transfer and further visualize the explanation results by actively and manipulatively synthesizing a series of new samples along defined paths. By introducing shortest paths searching strategy on the learned topology graph, we can efficiently obtain segmentation results by generating meaningful samples quickly for comparing.

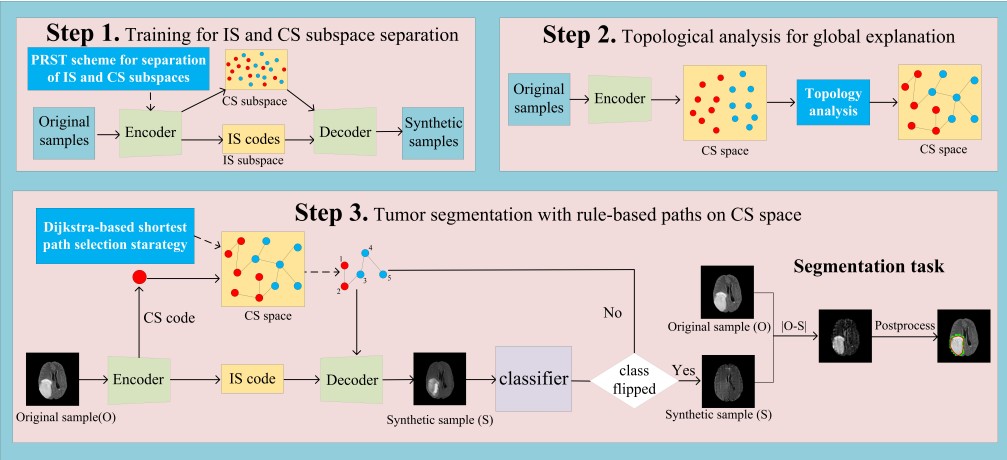

Figure 1: Overall framework of brain tumor segmentation.

## 2 METHODS

### 2.1 OVERALL EXPLAINABLE LEARNING FRAMEWORK FOR BRAIN TUMOR SEGMENTATION

Our proposed weakly supervised learning framework for brain tumor segmentation consists of three stages, as shown in Figure 1. In the first stage, we develop a symmetrical and cyclic generative adversarial network to effectively extract class-related features and create a unified manifold. This network also allows us to generate new samples with class reassignment by embedding class-related features, while preserving individual identity information. Moving on to the second stage, we utilize the trained encoder from the first stage to extract class-style (CS) codes from all original samples to be detected. We then perform topological data analysis on these extracted CS codes to generate a comprehensive topological graph, which represents the interrelationships and dependencies among the CS codes.

In the third stage, we follow a specific process to detect each original exemplar. Firstly, for one exemplar, we choose a goal counter exemplar (reference) and use the trained encoder from the first stage to extract their respective CS codes. These CS codes are then matched with the corresponding nodes on the topology graph, which represent the original and goal nodes. By utilizing matrix operations and the well-known Dijkstra algorithm, we determine the shortest path between these nodes. Along this path, we combine the original IS code with the center vectors of the nodes to synthesize a series of samples. This synthesis continues until the classifier predicts the synthetic sample as the flipped class. By comparing the original exemplar with the generated counter exemplar, we are able to effectively locate and segment the brain tumor. Furthermore, we employ topological data analysis to project class-related features onto a low-dimensional manifold. This allows us to manipulate the generation of new samples, altering their class attributes along multiple predefined rule-based paths. As a result, we are able to visualize and explain the global rules of the model across the entire dataset.

### 2.2 CLASS ASSOCIATION EMBEDDING FOR COUNTERFACTUAL GENERATION

Considering that it is hard to acquire labeled medical data in clinical scenes, we propose a novel approach for brain tumor segmentation that does not require expensive pixel-level annotation. We divide brain images into two categories: class $A$ for normal brain images and class $B$ for brain images with tumors. For brain images in class $B$, we generate new images that remove the tumor while preserving other tumor-unrelated features. This is achieved by embedding class-related features from normal samples. By comparing these generated images with the original ones, we can accurately segment the brain tumors. To ensure the accurate removal of tumors without compromising other important information in the brain image, such as structure, contour, and size, it is crucial to separate the class-related information from the class-unrelated (identity) information within the image. To address this challenge, we propose a symmetric cyclic generative adversarial network

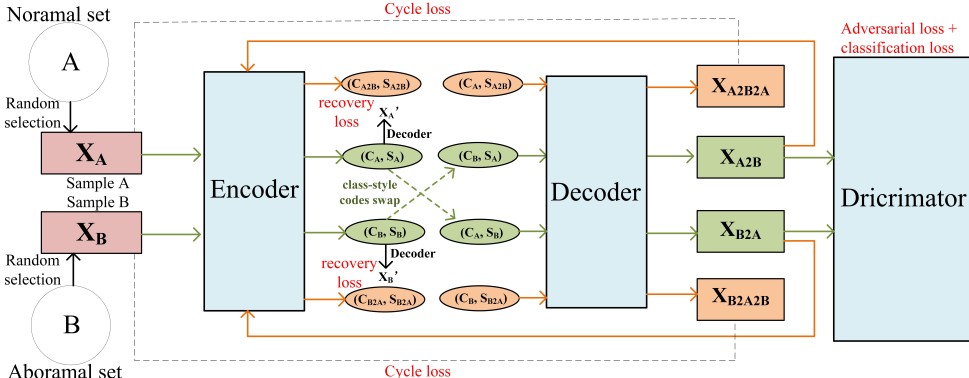

Figure 2: Symmetric cyclic adversarial network with paired random shuffle training scheme, while $C$ and $S$ refer to class-style and individual-style codes respectively.

architecture, as illustrated in Figure 2. This architecture consists of an encoder, a decoder, and a discriminator. The encoder incorporates two sub-modules responsible for encoding the class-related information and the individual identity information of the images. This results in the generation of class-style (CS) codes and individual-style (IS) codes, respectively. The decoder takes a combination of the IS code derived from the abnormal image and the CS code obtained from the normal image as input. It then generates a new image that effectively eliminates the tumor while preserving other important features. To separate class-related information from individual identity information well and generate real-looking images with successful class reassignment, a training approach called Paired Random Shuffle Training (PRST) with a series of loss functions is introduced. Specific training details are available in previous manuscript Xie et al. (2023). Overall, our approach offers a cost-effective solution for brain tumor segmentation by utilizing counterfactual generation and disentangling class-related and class-unrelated information.

## 2.3 TOPOLOGY ANALYSIS OF CS CODES FOR EXPLORING AND EXPLAINING GLOBAL RULE

The framework mentioned above compresses important class-related information into a low-dimensional space of 8 dimensions. To further study global rules and inter-class relationships, dimensionality reduction techniques like principal component analysis (PCA) or t-SNE Van der Maaten & Hinton (2008) can be used to visualize the latent representation. However, these methods may lead to information loss and have limited ability to explore correlations between samples and the overall structure of the dataset due to their linear combination representation. To overcome this limitation, we employ topological data analysis (TDA) Carlsson (2009); Offroy & Duponchel (2016) for feature visualization. TDA presents significant connections between samples and potential structural patterns of the data through a topological representation. This approach allows us to explore clustering, hierarchy, looping, isolated points, and other characteristics of the data, providing a more comprehensive understanding of the dataset. We utilize the Mapper algorithm Singh et al. (2007); Joshi & Joshi (2019) for topology analysis of the extracted class-style (CS) codes. The original CS codes (8 dimensions) are projected onto a one-dimensional axis using T-distribution or t-SNE processing. These codes are then divided into multiple overlapping covers. Within each cover, Density-Based Spatial Clustering of Applications with Noise (DBSCAN) Ester et al. (1996) is used for clustering analysis to identify samples belonging to the same class. These samples are represented by a single topological node, which is connected to other nodes that share samples. We apply the CS codes extracted from the trained counterfactual generation framework mentioned in section 2.2 for topology analysis. By using this method, we obtain a topological graph with numerous nodes and connecting lines, allowing us to analyze sample relations, explore and explain global rules within the learned manifold.

## 2.4 Designing Rule-based Paths on Topology Graph for Efficient and Explainable Segmentation

By utilizing the proposed framework and training scheme for counterfactual generation, we can effectively extract and consolidate class-related features into a unified and low-dimensional domain. The analysis of the extracted class-style (CS) codes reveals valuable insights into global rules and relationships among samples through the topological graph. This forms the foundation for performing the segmentation task effectively, with the aid of rule-based and efficient counterfactual generation techniques. To ensure accurate and efficient counterfactual generation, it is crucial to select meaningful CS codes for embedding. In this regard, we propose a selection strategy of CS codes that adopts rule-based and efficiency-based criteria, achieved through the design of guided and meaningful paths. For each sample to be segmented, we first select its nearest node (with the lowest Euclidean distance between the CS code of this sample and the center vector of CS codes involved in the node) as the initial node. We then randomly select another node as the goal node and establish the most efficient and shortest path between the initial and goal nodes based on the Dijkstra algorithm. To represent the topology graph, we calculate and derive an adjacency matrix initially. The entries in this matrix correspond to the Euclidean distances between the center vectors of the connected nodes (i.e., node $i$ and node $j$). For node pairs without a direct connection, the values are assigned as infinity. By utilizing this distance matrix and performing the Dijkstra operation, we can determine the shortest paths between every pair of nodes. Connections are then established between adjacent nodes along the defined path. This implies that the current generated sample will be linked with the sample synthesized in the previous stage if we choose CS codes from the center vectors of nodes along the defined path for embedding. As a result, the model behaviors and the disease development process can be explored, explained, and visualized in a guided and rule-based manner. Furthermore, by comparing synthetic samples with the original samples along these efficient and rule-based paths, we can achieve accurate and explainable segmentation of brain tumors.

## 3 Experiment

### 3.1 Dataset

We utilized two datasets, BraTS2020 bra and BraTS2021 Baid et al. (2021), for our experiments. These datasets were obtained from the Brain Tumor Segmentation challenge held at the 23rd and 24th International Conference on Medical Image Computing and Computer Assisted Intervention (MICCAI 2020 and 2021) respectively. The training datasets consist of 3D brain MRI scans accompanied by ground truth labels provided by expert board-certified neuro-radiologists. To generate the training data, we selected specific slices at coordinates 80, 82, 84, 86, 88, and 90 along the z-axis, resulting in the acquisition of six 2D brain images with six corresponding ground-truth masks for each patient from the training sets. For the BraTS2020 dataset, a total of 1,298 brain images were used in our experiments. Among these, 1,005 images contained tumors, while 293 images were normal. From this dataset, we randomly selected 704 abnormal brain images and 206 normal images for the training set. The remaining images formed the test set, which consisted of 301 abnormal images and 87 normal images. Regarding the BraTS2021 dataset, we randomly selected 3,828 abnormal brain images and 710 normal images for the training set. And the test set comprised 1,623 abnormal images and 302 normal images. In our experimental setup, we resize all input images to a size of $256 \times 256$ pixels. During training, we randomly apply horizontal flipping to the input images with a probability of 0.5. To update the network parameters, we utilize the Adam optimizer Kingma & Ba (2014) with an initial learning rate and weight decay set to 1e-4.

### 3.2 Overall Results of Brain Tumor Segmentation Task

We conducted experiments to validate the effectiveness of our proposed method for brain tumor segmentation, as described in Section 2. Our approach involved guided and meaningful generations based on rule-based and shortest paths, determined by the original and goal reference images, following the methodology outlined in Sections 2.1 and 2.4. By comparing the original samples with the synthetic samples, we obtained difference maps, which were then post-processed to generate the segmentation results. Figure 3 showcases some examples of the segmentation, demonstrating accurate detection of brain tumor regions using our method. To objectively evaluate the segmenta-

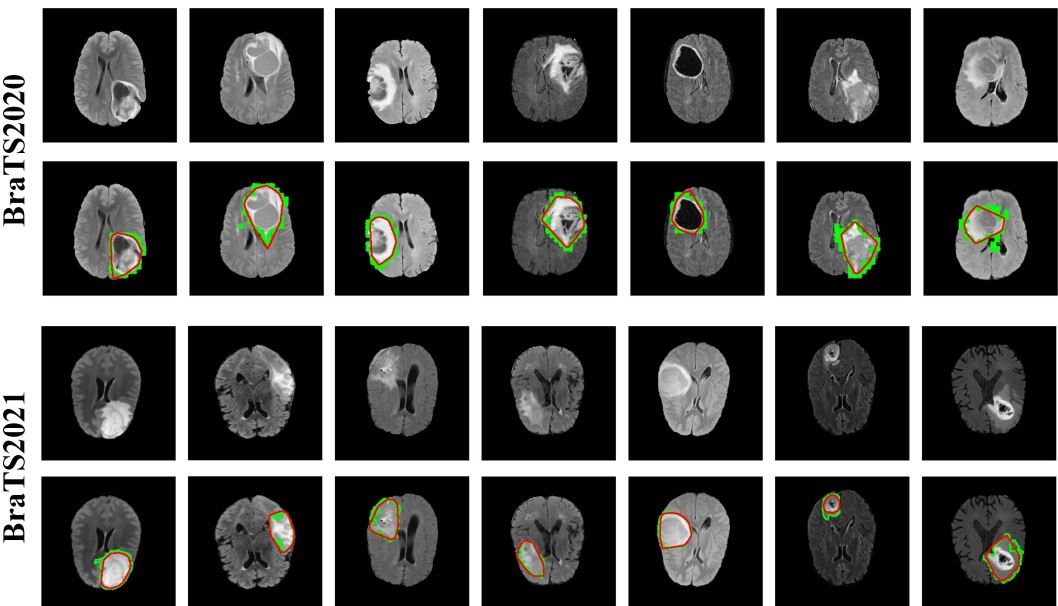

Figure 3: Cases of segmentation results using our proposed algorithm. Images in the first and the third rows are the cases to be detected, and corresponding segmentation results are presented in the second and fourth rows for BraTS2020 and BraTS2021 datasets respectively. The regions surrounded by the green lines are the groundtruth tumor regions, while the regions surrounded by the red lines are the predicted tumor regions using our proposed segmentation method.

tion performance, we calculated the Intersection over Union (IOU) and DICE values between the segmentation results and the corresponding ground-truth masks. On the BraTS2020 dataset, we achieved mean IOU and DICE values of 0.6373 and 0.7585, respectively, on the test set. For the BraTS2021 dataset, we obtained 0.5791 and 0.6915 on the IOU and DICE metrics, respectively. Importantly, our method does not require pixel-level annotations, which enhances its potential for widespread clinical application.

### 3.3 Topology Analysis Reveals Global and Explainable Class Transition Rules from Learned Manifolds

The main idea of this paper is to separate class-related features from class-unrelated features (identity features) to learn a low-dimensional manifold with discriminative class properties. This approach allows for accurate brain tumor segmentation and explainable artificial intelligence by generating new samples for comparison based on the learned unified class-related manifold. The rule-based and class-discriminative nature of the learned manifold is crucial. To assess the reliability of the learned class-related manifold, we performed topological data analysis on the class-style codes extracted from brain images in the test set using the trained encoder (as described in Section 2.3). The results of the topology analysis are shown in Figure 4, providing insights into the characteristics and structure of the learned manifold. The visual representations obtained through topology analysis clearly show a separation between normal and abnormal cases in the learned class-related manifold. It is worth noting that normal cases tend to cluster towards the left margin, while the proportion of abnormal cases gradually increases as the paths extend towards the right on the topology graph. This alignment with the progression of disease development provides valuable and explainable insights. Additionally, the topology graph reveals the intricate relationships between groups, samples, and classes, thanks to the numerous connection lines established. This rich connectivity allows for comprehensive analysis of these relationships and exploration and explanation of global rules for the entire dataset. This is particularly advantageous in clinical settings where doctors can benefit from detailed insights.

To demonstrate the explainable and rule-based nature of the class-related manifold more clearly, we randomly chose two normal examples and extracted their individual-style (IS) codes using the

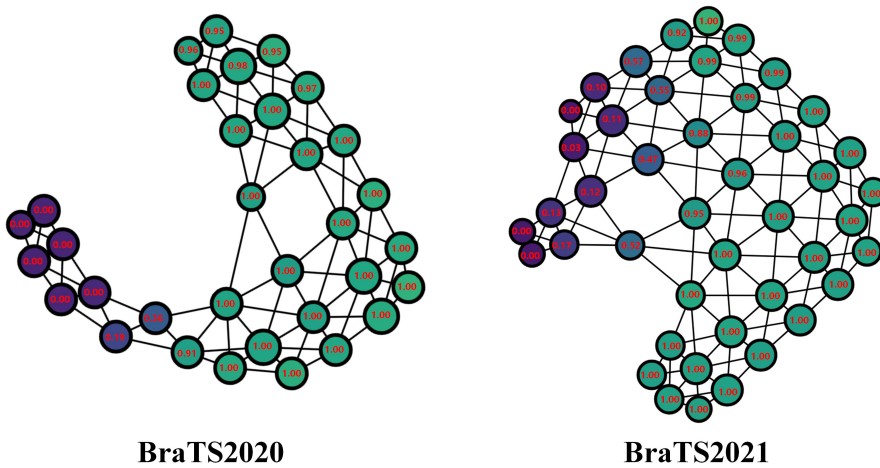

BraTS2020                    BraTS2021

Figure 4: The results of topology analysis of learned class-related manifolds on BraTS2020 and BraTS2021 datasets. In the topology graph, values with red font inside the nodes refer to the ratios of the abnormal cases involved in these nodes.

trained encoder. Then, we defined two distinct paths randomly, as shown in Figure 5, to generate a series of new images. These images were created by combining the original IS codes with the center vectors of the class-style (CS) codes associated with each node along the defined paths. The generated synthetic samples not only retain the original identity features but also show a noticeable trend of tumor development and the emergence of more pronounced disease-related characteristics along the defined paths. This further confirms the successful acquisition of rule-based and explainable knowledge within our manifold. Additionally, this capability allows us to accurately and explainably segment tumors by embedding meaningful CS codes along with defined rule-based paths that are consistent with the medical knowledge and generating samples that demonstrate successful class reassignment.

### 3.4 Effectiveness of Paths Design Strategy Using Topology Graph for Brain Tumor Segmentation

To showcase the efficient implementation of precise segmentation, we conducted a compared study by embedding class-style (CS) codes along paths generated through t-SNE analysis. We firstly projected all class-related codes of the test images into the t-SNE manifold. Due to the absence of connections between samples, we could only perform sampling between original and goal samples to generate new class-related codes for embedding for image generation. We sampled CS codes at regular intervals of 0.1d (where $d$ represents the Euclidean distance between the CS codes of the original and reference samples) from a linear path defined by the CS codes of the original and reference samples in the t-SNE manifold. In the BraTS2020 dataset, the resulting mean IOU and DICE values achieved in the test set were 0.4516 (reduced by 29.14%) and 0.5752 (reduced by 24.17%) respectively. In the BraTS2021 dataset, the method that designs paths using linear sampling on the t-SNE manifold achieved an IOU of 0.5372 (reduced by 7.24%) and a DICE metric of 0.6482 (reduced by 6.26%).

This comparison demonstrates the effectiveness of our proposed path design strategy using topology graphs for generating CS codes and synthesizing new samples for the segmentation task. It is worth noting that the number of operations required to generate samples in the t-SNE manifold were 2302 and 15412 in the BraTS2020 and BraTS2021 datasets. In contrast, our proposed shortest path design method using the topology manifold only required 1305 and 7645 operations (reduced by 43.31% and 50.40% respectively) for the BraTS2020 and BraTS2021 datasets respectively. This observation further supports the high efficiency and reliability of our proposed method in generating meaningful CS codes and synthesizing new samples for accurate brain tumor segmentation.

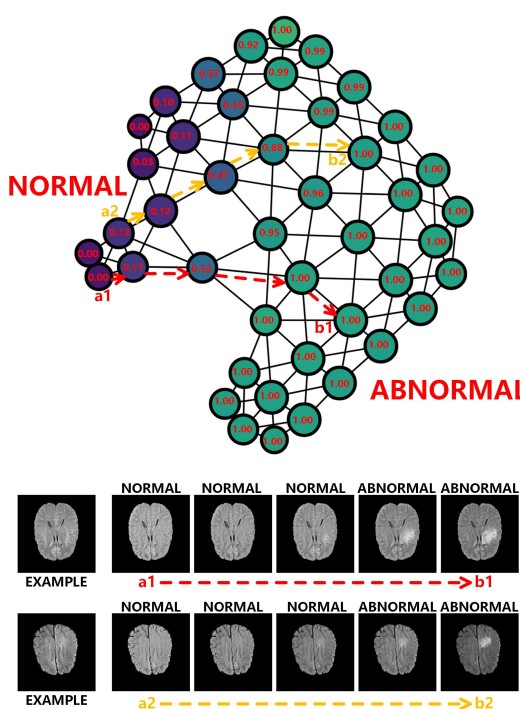

Figure 5: Generated cases (located on the right of the EXAMPLE) along two defined paths ($a1$ to $b1$ and $a2$ to $b2$). The individual-style codes of the EXAMPLE are extracted for combinations with the center vectors of the class-style codes involved in each node along the defined paths. The predicted classes by the external classifier are presented above the synthetic cases.

Table 1: Mean IOU and DICE values on BraTS2020 and BraTS2021 test sets using TSCAM, ICAM, SCAM, Bridging, LAGAN and our proposed algorithm.

| Methods | $TSCAM$ | $ICAM$ | $SCAM$ | $Bridging$ | $LAGAN$ | $Ours$ |
|---|---|---|---|---|---|---|
| IOU (BraTS2020) | 0.5810 | 0.5729 | 0.5378 | 0.4549 | 0.4214 | **0.6373** |
| DICE (BraTS2020) | 0.6972 | 0.7138 | 0.6657 | 0.6180 | 0.5556 | **0.7585** |
| IOU (BraTS2021) | 0.5497 | 0.2814 | 0.5375 | 0.2158 | 0.4220 | **0.5791** |
| DICE (BraTS2021) | 0.6540 | 0.4002 | 0.6495 | 0.3470 | 0.5420 | **0.6915** |

### 3.5 COMPARISON WITH OTHER ALGORITHMS FOR BRAIN TUMOR SEGMENTATION

To demonstrate the superiority of our method in the brain tumor segmentation task, we compared our proposed segmentation method with other existing weakly supervised learning algorithms (TSCAM Yao et al. (2022), ICAM Bass et al. (2022), SCAM Bai et al. (2022), Bridging Kim et al. (2022) and LAGAN Tao et al. (2023)). Figure 1 presents several segmented cases using both other segmentation algorithms and our method, clearly illustrating the better segmentation results achieved with our approach. We further substantiated the superiority of our method by comparing it with other algorithms using the IOU and DICE metrics. The results in Table 1 indicate higher mean IOU and DICE values on both datasets using our proposed method, providing additional evidence that our approach outperforms others in the brain tumor segmentation task.

## 4 CONCLUSION

The computer-aided diagnosis of brain tumors offers significant advantages for both patients and medical professionals. However, current approaches for brain tumor detection often rely on expensive pixel-level annotations and lack explainability. In this study, we aim to address these common

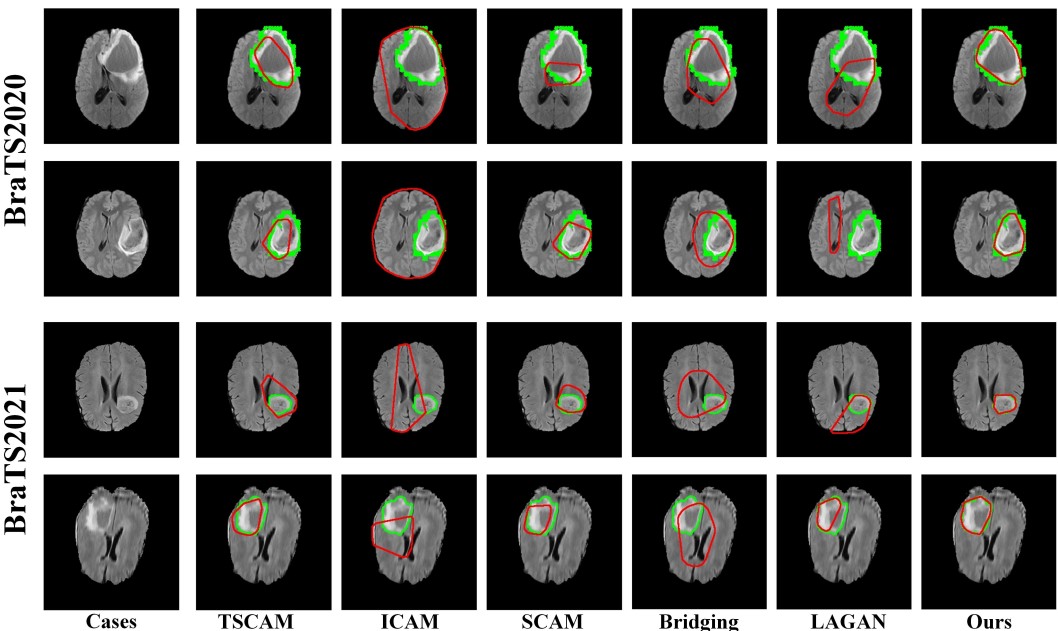

Figure 6: Cases of segmentation results using TSCAM, ICAM, SCAM, Bridging, LAGAN and our proposed algorithm. The regions surrounded by the green lines are the groundtruth tumor regions, while the regions surrounded by the red lines are the predicted tumor regions.

challenges by proposing a weakly supervised learning framework that eliminates the need for pixel-level annotations in brain tumor segmentation and incorporates global explainable learning. Our proposed model effectively learns a unified manifold that captures class-related information while maintaining global explainability and class discriminability. By performing topological data analysis on learned class-related features and designing meaningful paths on the topological graphs, we synthesize a series of new meaningful samples along rule-based paths, enabling the exploration of global class transition rules and global explanation of black-box model behaviors. This approach facilitates task such as brain tumor segmentation by actively manipulating generation for comparison, providing valuable insights that are difficult to achieve through conventional methods and holding potential for clinical implementation. Compared to existing methods, our proposed approach achieves superior results in brain tumor segmentation, highlighting its effectiveness and advantages in this context.

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
