# OpenReview forum: "A Weakly Supervised and Globally Explainable Learning Framework for Brain Tumor Segmentation"
_ICLR.cc/2024/Conference — ICLR 2024 Conference Withdrawn Submission_

### Official Review · Reviewer_4gaV · 2023-10-30

**Soundness:** 2 fair
**Presentation:** 3 good
**Contribution:** 2 fair
**Rating:** 3
**Confidence:** 4

**Summary:**

In this work, the authors proposed a counterfactual generation framework for the brain tumor segmentation task. This work not only performs self-supervised segmentation accurately without relying on pixel-level annotations but also provides explainability, which is lacking in most of the deep learning models. The proposed method efficiently segregates class-related features from class-unrelated features within the samples. It then generates new samples that maintain identity features while modifying class attributes by embedding diverse class-related features. To achieve this, authors employ topological data analysis to project the extracted class-related features onto a globally interpretable class-related manifold.

**Strengths:**

1.	The paper reads well and easy to understand.
2.	The idea of extracting class-style codes to get the insights into global rules and creating relationships among topological structure adds some novelty.

**Weaknesses:**

1.	Methodology section lacks math behind the study, equations and optimization strategy.
2.	Missing some SOTA during evaluation (eg, Capturing implicit hierarchical structure in 3D biomedical images with self-supervised hyperbolic representations and other self-supervised methods) and fully supervised methods.
3.	Missing ablation study on design choices (eg, adding shortest path design).

**Questions:**

1.	It’s interesting to see how this can be extended for multi-class segmentation. In most tumor cases, hardest region of interest to extract is enhanced tumor region. So how can you extend your approach with multi-class style codes and multiple rule-based approach?
2.	What is the computational complexity of the proposed method?

---

### Official Review · Reviewer_Er7v · 2023-10-31

**Soundness:** 2 fair
**Presentation:** 2 fair
**Contribution:** 2 fair
**Rating:** 3
**Confidence:** 3

**Summary:**

In healthcare, brain tumors are a significant clinical issue that affects many patients. Machine-based tumor segmentation can assist doctors in diagnosis and treatment planning. Deep learning techniques offer potential for feature representation in brain tumor segmentation but often require a large number of samples with pixel-level annotations, and they lack interpretability. The paper addresses these challenges by introducing a framework that separates class-related features from class-unrelated ones. This approach reduces the need for pixel-level annotations, making it more practical for clinical settings. To improve interpretability, the authors use topological data analysis to project class-related features into a globally explainable manifold. This adds a layer of explanation to the model. The method is validated on two distinct datasets and shows improved performance in brain tumor segmentation, contributing to both machine learning and medical imaging.

**Strengths:**

- The paper focuses on brain tumor segmentation, a critical issue in medical imaging and diagnostics. Effective segmentation can significantly aid in the treatment planning and monitoring of brain tumors, making this a high-impact area of research.

- The paper acknowledges the challenges of requiring detailed, pixel-level annotations for medical images, especially in clinical settings. By aiming for a weakly supervised approach, the paper shows an understanding of the practical limitations and costs involved in medical image annotation.

- The title suggests an emphasis on creating a "globally explainable" machine learning framework. Explainability is particularly crucial in medical applications, where understanding the model's decisions can influence clinical decisions and patient trust. Focus on this aspect is a strength

**Weaknesses:**

- The paper does not talk about what it means to have a unified manifold in section 2.2 and could have done a better job outlining some insights here.
- This paper indexes on explainability but does not really outilne why their method has explainability and additionally the claim that topological analysis lends interpretability is not established.
- The paper does not provide a detailed discussion on the limitations or potential challenges of the proposed weakly supervised learning framework for brain tumor segmentation.
- They experiment with
- The paper does not mention any comparative analysis or evaluation of the proposed method against existing state-of-the-art approaches for brain tumor segmentation.
- The paper lacks information on the size and diversity of the datasets used for evaluation, which could affect the generalizability of the proposed method.
- The paper does not discuss the computational complexity or efficiency of the proposed framework, which could be important considerations for practical implementation.
- The paper does not provide insights into the potential impact or implications of the proposed method in real-world clinical settings, such as its integration with existing medical imaging systems or its usability by healthcare professionals

**Questions:**

- What are the reasons that the method is inherently explainable?
- How do you ensure that the space allows for linear interpolation in generating new samples?
- Why do you choose to create a graph when you could have used distance based algorithms to samples close in the euclidian space? What are the added advantages of this approach?
- How do you handle errors from the generation network - if you would like it to generate images from class A but it generates images from class B, how do you handle the error?

---

### Official Review · Reviewer_Bw47 · 2023-10-31

**Soundness:** 2 fair
**Presentation:** 3 good
**Contribution:** 1 poor
**Rating:** 3
**Confidence:** 4

**Summary:**

This paper presents a counterfactual generation framework that not only achieves exceptional performance in brain tumor segmentation without the need for pixel-level annotations but also provides explainability. The experimental results on two datasets are conducted to validate the effectiveness of the proposed framework.

**Strengths:**

(1) The overall structure is clear.
(2) The proposed framework can effectively separate class-related features from class-unrelated features of the samples, and generate new samples that preserve identity features while altering class attributes.
(3) The experiments were conducted on two public brain tumor datasets to validate the effectiveness of the proposed model.

**Weaknesses:**

(1) The overall contribution is limited. The core is this study aims to generate some normal samples, and then extract the tumor regions by using the generated samples.
(2) The tumor regions are identified by performing a comparison between the original abnormal images and generated normal samples. However, if the generated samples are not accurate, the identified tumor regions are affected.
(3) The paper designs weakly-supervised segmentation methods, but they do not compare it with other SOTA weakly-supervised segmentation models.

**Questions:**

(1) The overall contribution is limited. The core is this study aims to generate some normal samples, and then extract the tumor regions by using the generated samples.
(2) The tumor regions are identified by performing a comparison between the original abnormal images and generated normal samples. However, if the generated samples are not accurate, the identified tumor regions are affected.
(3) The paper designs weakly-supervised segmentation methods, but they do not compare it with other SOTA weakly-supervised segmentation models.

---

### Official Review · Reviewer_naJ9 · 2023-10-31

**Soundness:** 2 fair
**Presentation:** 2 fair
**Contribution:** 2 fair
**Rating:** 3
**Confidence:** 4

**Summary:**

The authors propose an interesting approach for a weekly supervised semantic segmentation framework. The proposed approach disentangles class-specific features from others, followed by identity-preserving counterfactual generation. The framework seems like a contrastive learning setup for a given pair of normal and diseased images for semantic segmentation. The steps mainly involve i) identifying class-specific features from others, ii) identity preserving counterfactual generation converting diseased to normal brain, (iii) pixel-wise L1 difference estimation, followed by (iv) post-processing to extract boundaries.

**Strengths:**

- The proposed method is innovative, combining multiple approaches, lifting the need for pixel-wise labels for semantic segmentation tasks
- Nice illustrative figures explaining the proposed method
- The proposed approach attempts to learn interpretable rules based on class-specific feature topology

**Weaknesses:**

- The proposed approach only works on segmenting the whole tumour, which seems limited.
- The interpretable rule from CS topology analysis is not that clear. Does the approach have a global CS topology graph from which the features are selected at every iteration of counterfactual generation? If so, how is this done?
- The reported performance on the whole tumour is about 15-20% lower than the state-of-the-art model (based on fully supervised learning); it's unclear how the method would scale to other classes, like tumour core and enhancing tumour
- I fail to understand how the method extends to a multi-class semantic segmentation setup. Do you need more than two images in a pair? Do you need to generate all possible counterfactuals?
- Analysis of the approach on other datasets would be really helpful
- Similar approaches like in anomaly detection literature are not discussed; refer [1, 2]
- Presentation concerns: move all the related works into the related work section rather than including them in the methods section, which makes it hard to follow the contribution.


[1] Chaitanya, K., Erdil, E., Karani, N. and Konukoglu, E., 2020. Contrastive learning of global and local features for medical image segmentation with limited annotations. _Advances in neural information processing systems_, _33_, pp.12546-12558.

[2] Wolleb, J., Bieder, F., Sandkühler, R. and Cattin, P.C., 2022, September. Diffusion models for medical anomaly detection. In _International Conference on Medical image computing and computer-assisted intervention_ (pp. 35-45). Cham: Springer Nature Switzerland.

**Questions:**

Please refer to weakness section